# Comparative Analysis of Transcriptomics and Metabolomics Reveals Defense Mechanisms in Melon Cultivars against *Pseudoperonospora cubensis* Infection

**DOI:** 10.3390/ijms242417552

**Published:** 2023-12-16

**Authors:** Yueming Ling, Xianpeng Xiong, Wenli Yang, Bin Liu, Yue Shen, Lirong Xu, Fuyuan Lu, Meihua Li, Yangdong Guo, Xuejun Zhang

**Affiliations:** 1Hami-Melon Research Center, Xinjiang Academy of Agricultural Sciences, Urumqi 830091, China; lingyueming@xaas.ac.cn (Y.L.); yangwenli@xaas.ac.cn (W.Y.); liubincau@163.com (B.L.); shenyue052442@163.com (Y.S.); jxy5179@zohomail.cn (L.X.); limeihua@xaas.ac.cn (M.L.); 2Shenzhen Branch, Guangdong Laboratory of Lingnan Modern Agriculture, Key Laboratory of Synthetic Biology, Ministry of Agriculture and Rural Affairs, Agricultural Genomics Institute at Shenzhen, Chinese Academy of Agricultural Sciences, Shenzhen 518120, China; xiongxianpeng@caas.cn; 3College of Horticulture, Xinjiang Agricultural University, Urumqi 830091, China; 4College of Agriculture, Shihezi University, Shihezi 832003, China; 14769674215@163.com; 5College of Horticulture, China Agricultural University, Beijing 100193, China; 6Hainan Sanya Experimental Center for Crop Breeding, Xinjiang Academy of Agricultural Sciences, Sanya 572019, China

**Keywords:** melon, downy mildew, transcriptomic, metabolomic, flavonoid, lignin

## Abstract

Melon (*Cucumis melo* L.) represents an agriculturally significant horticultural crop that is widely grown for its flavorful fruits. Downy mildew (DM), a pervasive foliar disease, poses a significant threat to global melon production. Although several quantitative trait loci related to DM resistance have been identified, the comprehensive genetic underpinnings of this resistance remain largely uncharted. In this study, we utilized integrative transcriptomics and metabolomics approaches to identify potential resistance-associated genes and delineate the strategies involved in the defense against DM in two melon cultivars: the resistant ‘PI442177′ (‘K10-1′) and the susceptible ‘Huangdanzi’ (‘K10-9′), post-*P. cubensis* infection. Even in the absence of the pathogen, there were distinctive differentially expressed genes (DEGs) between ‘K10-1′ and ‘K10-9′. When *P. cubensis* was infected, certain genes, including flavin-containing monooxygenase (FMO), receptor-like protein kinase FERONIA (FER), and the HD-ZIP transcription factor member, AtHB7, displayed pronounced expression differences between the cultivars. Notably, our data suggest that following *P. cubensis* infection, both cultivars suppressed flavonoid biosynthesis via the down-regulation of associated genes whilst concurrently promoting lignin production. The complex interplay of transcriptomic and metabolic responses elucidated by this study provides foundational insights into melon’s defense mechanisms against DM. The robust resilience of ‘K10-1′ to DM is attributed to the synergistic interaction of its inherent transcriptomic and metabolic reactions.

## 1. Introduction

Melon (*Cucumis melo* L.) belongs to the Cucurbitaceae family and is renowned for its abundance of carbohydrates and nutrients. Its significant horticultural and economic value has made it a prominent crop globally [1]. Global melon cultivation and production surged to 28.6176 million tons and 1.0774 million hectares, marking a 4% and 7% increase over the preceding five years, respectively [2]. Notwithstanding this growth, global melon yield faces significant threats from downy mildew (DM), a formidable disease instigated by *Pseudoperonospora cubensis* (*P. cubensis*), an oomycete class member. *P. cubensis*, a biotrophic plant fungus, belongs taxonomically to stramenopiles (kingdom), oomycota (phylum), oomycetes (class), peronosporales (order), peronosporaceae (family), and pseudoperonospora (genus). This pathogen not only naturally infects melon but also affects over 20 genera and 60 species within the Cucurbitaceae family, including cucumbers, watermelons, pumpkins, luffa, winter squash, and melon [1,3]. The fungus’ spores, disseminated extensively via air currents, impact a broad spectrum of host plants and have exhibited a swift resistance to fungicides, amplifying their global threat [4]. The prevailing reliance on chemical agents for controlling melon DM has triggered serious environmental and health concerns, alongside bolstering pathogenic resistance to these chemicals [5,6,7]. Consequently, unearthing disease-resistant genes in melons and advancing molecular breeding techniques emerge as crucial strategies. Such approaches not only augment melon production in terms of both volume and quality but also contribute significantly to the melon industry’s sustainable development.

In recent years, there has been an increasing emphasis among researchers on leveraging multi-omics data, encompassing metabolomics and transcriptomics, to elucidate metabolic pathways and molecular regulatory networks integral to plant disease resistance mechanisms [8,9,10,11]. This holistic approach not only aids in pinpointing pivotal genes but also provides a foundational framework for breeding disease-resistant varieties. For example, by integrating metabolomics and transcriptomics, Jeon et al. identified a gene cluster associated with tomato hydroxycinnamic acid biosynthesis, underscoring its role in enhancing tomato resistance to pathogens [8]. Similarly, Su et al., utilizing the LC-MS/MS platform for metabolomic analysis and employing the Pearson correlation coefficient for data assessment, deduced that *Fusarium graminearum* infection instigated the wheat flavonoid biosynthesis pathway [9]. Li et al. investigated the transcriptomic and metabolomic shifts in both disease-resistant and susceptible varieties post-Fusarium zanthoxyli infection [10]. Their correlation analysis between gene expression and metabolite synthesis revealed a pronounced role of flavonoid metabolism in *Zanthoxylum bungeanum*’s defense against stem canker disease. Xu et al. further expanded the multi-omics approach by amalgamating transcriptomic, proteomic, and metabolomic data, unveiling that jasmonic acid signaling modulated phenolic acid accumulation in barley [11]. This accumulation subsequently bolstered barley’s defense against powdery mildew via H3K27me3 demethylation and the activation of defense-centric gene expression. Nonetheless, in spite of the in-depth investigations on disease-centric metabolism and key regulatory genes across diverse plants, a knowledge gap concerning genes modulating metabolite accumulation linked with resistance to DM remains, especially in economically vital crops, like sweet melon.

Research on DM in melon predominantly emphasizes disease resistance prevention, control measures, genetic analysis, and the pinpointing and characterization of resistance genes. To date, the recognized DM-resistant genes in melon encompass *Pc-1* (*AT1*, an incomplete dominant gene) and *Pc-2* (*AT2*, an incomplete dominant gene), *Pc-3* (a single dominant gene), *Pc-4* (an incomplete dominant gene), and *Pc-5* (either a single dominant gene or heterozygous) [12]. Subsequent investigations validated that *Pc-1* and *Pc-2* correspond to the genes *AT1* and *AT2*, respectively, both of which are linked to serineglyoxylate aminotransferase (SGT). Remarkably, both *AT1* and *AT2* manifest elevated expression levels in DM-resistant melon lines compared to their DM-susceptible counterparts. Furthermore, the overexpression of either gene can bestow DM resistance upon susceptible melon plants, implying that resistance might be attained through adjustments in enzyme activity that instigate defense responses [12,13]. Although *Pc-3*, *Pc-4*, and *Pc-5* are known to confer resistance to DM, their specific genetic underpinnings and operative mechanisms remain elusive. In recent advancements, four salient QTLs have been detected in the DM-resistant strain MR-1: *qPcub-10.1*, *qPcub-8.2*, *qPcub-10.3-10.4*, and *qPcub-8.3* [14,15]. Intriguingly, within the *qPcub-10.1* localization interval, a putative gene, denoted as MLO-like protein, has been identified, bearing two synonymous SNPs [15].

Recent integrative transcriptomic and metabolomic analyses have elucidated a notable correlation between flavonoid biosynthesis and the peel color of sweet melon [16]. In a similar vein, Nagashima et al. harnessed both metabolomic and transcriptomic tools to discern alterations in volatile compounds during the fruit maturation process in two Hami melon cultivars [17]. Nevertheless, to date, no investigations have delved into the regulatory intricacies of melon DM resistance using this combined omic approach. Furthermore, a distinct gap in our understanding persists regarding the modulation of transcriptomics and metabolomics in melon leaves, especially pertaining to resistance pathways under varied nutritional, environmental, and stress contexts within certain species [18,19]. In the present study, we juxtaposed the foundational transcriptome and metabolome of two divergent DM-resistant melon varieties prior to *P. cubensis* infection. Subsequently, we gauged the differential responses of these cultivars at the transcriptomic and metabolomic strata following the initial *P. cubensis* invasion. Comprehensive analyses, encompassing transcriptomics, metabolomics, and the quantification of flavonoids and lignin, illuminated the defensive tactics employed by distinct melon varieties against *P. cubensis*, elucidating the underpinnings of their differential susceptibilities to DM.

## 2. Results

### 2.1. Phenotypic Response of Melon to P. cubensis Inoculation

To assess the resistance of ‘K10-1′ and ‘K10-9′ to downy mildew (DM), inoculation experiments were performed during both seedling and mature stages. During the seedling inoculation trial, water-soaked spots manifested on ‘K10-9′ three days post-inoculation. As the disease progressed, these spots expanded into yellow-brown angular lesions accompanied by gray-black fungal growth by the seventh day. This progression culminated in pronounced leaf yellowing and even wilting (Figure 1a). Conversely, ‘K10-1′ only exhibited an allergic response on the fifth day post-*P. cubensis* inoculation. In the mature stage inoculation, no overt phenotypic alterations were observed in ‘K10-1′ fifteen days post-inoculation. However, ‘K10-9′ displayed significant leaf yellowing and wilting at the same time point (Figure 1b). Across both growth stages, ‘K10-1′ consistently registered a disease severity score markedly lower than that of ‘K10-9′ (Figure 1c). Collectively, these observations underscore substantial disparities in DM resistance between ‘K10-1′ and ‘K10-9′, rendering them suitable candidates for exploring the underlying resistance mechanisms in melon against DM.

### 2.2. Transcriptomic Variability in Melon Resistance and Susceptibility to P. cubensis

To elucidate the molecular underpinnings of melon resistance to DM at both transcriptomic and metabolic dimensions, transcriptome sequencing was conducted on leaf samples from ‘K10-1′ and ‘K10-9′ subjected to *P. cubensis* infection for 48 h. Subsequent to meticulous filtering and quality assessment, an average of 10.74 Gb of high-quality reads were generated. Notably, over 94.84% of these reads were uniquely aligned with the melon reference genome (Appendix A). The proportion of Q20 bases spanned 95.99% to 97.37%, Q30 bases ranged between 89.54% and 92.44%, and the mean GC content stood at 45.03% (Appendix A). The robust consistency across the three independent biological samples was underscored by the Pearson correlation coefficient analysis, which yielded a coefficient r ≥ 0.8 (Appendix A). Principal Component Analysis (PCA) revealed a tight clustering among the biological replicates, with PCA1 and PCA2 adeptly segregating inoculated from non-inoculated samples and differentiating between the cultivars (Figure 2a). This pattern underscores the distinct gene expression profiles across the cultivars in the context of *P. cubensis* exposure.

Given the established role of basal transcription in modulating disease resistance responses [20,21], we pursued transcriptomic analyses of the ‘K10-1′ and ‘K10-9′ melon cultivars in the absence of *P. cubensis* inoculation. Without *P. cubensis* exposure, a distinction of 1551 differentially expressed genes (DEGs) was observed between the two cultivars, adhering to criteria |log2Fold Change| ≥ 1 and FDR < 0.05. Specifically, 651 genes manifested heightened expression in ‘K10-1′ relative to ‘K10-9′, whereas the expression of 900 genes was diminished in ‘K10-1′ (Figure 2b). Upon inoculation with *P. cubensis*, the differential gene count escalated to 2142 between the cultivars. In this context, 775 genes showcased amplified expression in ‘K10-1′, in contrast to 1367 genes that registered attenuated expression in the same cultivar (Figure 2b). Independent of *P. cubensis* exposure, a recurring pattern was discerned, wherein 293 genes were consistently up-regulated and 447 genes were down-regulated in ‘K10-1′ relative to ‘K10-9′ (Figure 2c and Appendix A).

Gene ontology (GO) enrichment analysis revealed that 293 genes consistently exhibited heightened expression in ‘K10-1′, and significant enrichment was observed in pathways associated with the defense response to fungus (GO:0050832), interaction with symbionts (GO:0051702), and jasmonic acid hydrolase activity (GO:0120091) (Figure 2d). Contrastingly, 447 genes exhibiting lower expression levels in ‘K10-1′ compared to ‘K10-9′, whether inoculated or not with *P. cubensis*, were associated with functions including endopeptidase inhibitor activity (GO:0004866), peptidase inhibitor activity (GO:0030414), and phospholipid binding (GO:0005543) (Figure 2d). In alignment with the transcriptomic data (Figure 2e, Appendix A), qRT-PCR results indicated that genes associated with the defense response to the fungus pathway, specifically the Flavin-containing monooxygenase (FOM, *MELO3C013562.2*) and receptor-like protein kinase FERONIA (FER, *MELO3C026311.2*), manifested amplified expression in ‘K10-1′ relative to ‘K10-9′, both prior to and following *P. cubensis* inoculation (Appendix A). Prior research has underscored the integral regulatory functions of these two genes in orchestrating plant systemic disease resistance [22,23].

### 2.3. Basal Metabolite Variations between P. cubensis-Resistant and Susceptible Melon Cultivars

In an effort to pinpoint metabolites underpinning DM resistance in melon, we undertook a metabolomic analysis of samples identical to those utilized for transcriptomic studies. The PCA plot manifested tight clustering patterns for biological replicates within respective groups, underscoring robust intra-group replicative consistency (Figure 3a). PC1 and PC2 collectively accounted for 50% of the observed variance, effectively distinguishing among the cultivars and their respective inoculation statuses. Echoing the transcriptomic insights, PCA delineated intrinsic metabolomic distinctions between the ‘K10-1′ and ‘K10-9′ cultivars, potentially indicative of their differential disease resilience.

In the absence of *P. cubensis* inoculation, the relative abundance of 81 metabolites was accentuated in ‘K10-1′ compared to ‘K10-9′, whereas ‘K10-1′ displayed diminished levels of 126 metabolites relative to ‘K10-9′ (Figure 3b). Post-*P. cubensis* exposure, ‘K10-1′ surpassed ‘K10-9′ in the relative content of 60 metabolites but was outmatched in 132 metabolites (Figure 3b). Regardless of *P. cubensis* inoculation, ‘K10-1′ consistently exhibited higher levels of 35 metabolites compared to ‘K10-9′, while 99 metabolites consistently displayed lower levels in ‘K10-1′, comprising 57.14% and 71.72% of the flavonoid proportions, respectively (Figure 3c; Appendix A). A KEGG analysis of these differentially abundant metabolites (DAMs) unveiled their functional implications. Notably, the 35 metabolites with heightened content in ‘K10-1′ prominently featured in the flavonoid biosynthesis pathway (ko00941), an avenue closely tied to plant disease defense mechanisms (Figure 3d). Conversely, within ‘K10-1′, the 99 metabolites with reduced content predominantly clustered within the flavone and flavonol biosynthesis pathway (ko00944) (Figure 3d). Collectively, these data spotlight a pronounced and consistent variance in the relative abundance of 91 flavonoid compounds between ‘K10-1′ and ‘K10-9′, irrespective of *P. cubensis* exposure.

### 2.4. Differential Transcriptional Response in the Two Cultivars Following P. cubensis Inoculation

To elucidate the divergent response mechanisms of the resistant cultivars to *P. cubensis* infection, we assessed the transcriptional differences between ‘K10-1′ and ‘K10-9′ at 48 h post-inoculation relative to the non-inoculated controls. Using the thresholds |log2Fold Change| ≥ 1 and FDR < 0.05, we discerned 2345 up-regulated and 2520 down-regulated genes in ‘K10-1′. In contrast, ‘K10-9′ exhibited 2321 up-regulated and 2509 down-regulated genes (Figure 4a). Subsequent to *P. cubensis* exposure, both cultivars shared 1586 up-regulated and 1705 down-regulated genes (Figure 4b). GO enrichment analysis pinpointed that the gene with an elevated expression in both cultivars predominantly participated in RNA modification (GO:0009451), chloroplast RNA modification (GO:1900865), and rhythmic process (GO:0048511). In contrast, commonly down-regulated genes were associated with photosynthesis (GO:0015979), photosystem (GO:0009521), and plastid membrane (GO:0042170) processes (Appendix A).

Post-infection, ‘K10-1′ registered significant up-regulation in 759 genes and down-regulation in 815 genes. ‘K10-9′ demonstrated significant up-regulation in 735 genes and down-regulation in 804 genes (Figure 4b). GO enrichment analysis highlighted genes exclusive to ‘K10-1′ with an elevated expression being affiliated with DNA repair (GO:0006281), cellular response to DNA damage stimulus (GO:0006974), and recombinational repair (GO:0000725). Conversely, genes with a suppressed expression in ‘K10-1′ were linked to phloem development (GO:0010088), phloem or xylem histogenesis (GO:0010087), and the regulation of cell wall organization or biogenesis (GO:1903338) (Figure 4c). Genes with a diminished expression in ‘K10-9′ were predominantly involved in the mitotic cell cycle (GO:0000278), polymeric cytoskeletal fiber (GO:0099513), and microtubule cytoskeleton (GO:0015630), while those with an augmented expression participated in response to water deprivation (GO:0009414), water response (GO:0009415), and salicylic acid response (GO:0009751) pathways (Figure 4d).

Previous research pinpointed the instrumental role of the homeobox-leucine zipper protein ATHB-7 (MELO3C005992.2), EID1-like F-box protein 3 (MELO3C014347.2), and LEAF RUST 10 DISEASE-RESISTANCE LOCUS RECEPTOR-LIKE PROTEIN KINASE-like 1.2 (MELO3C035551.2) in drought stress responses [24,25,26] (Appendix A). Our experiment observations corroborate that ‘K10-9′ surpassed ‘K10-1′ in drought tolerance upon exposure to drought conditions (Appendix A). Furthermore, pivotal genes like transcription factor TGA2-like (MELO3C012485.2), WRKY transcription factor (MELO3C009097.2), and WRKY transcription factor 70 (MELO3C020166.2) saw marked up-regulation in the salicylic acid response pathway in ‘K10-9′ post-*P. cubensis* exposure (Appendix A). This points to the triggering of the SA signaling pathway in ‘K10-9′. Contrastingly, in ‘K10-1′, these genes did not display notable transcriptional shifts post-inoculation, with some even recording significantly lower expressions compared to ‘K10-9′ (Appendix A).

### 2.5. Differential Metabolic Response in the Two Cultivars after P. cubensis Inoculation

To elucidate the specific metabolites contributing to DM response, we examined the metabolic variations between the two cultivars after *P. cubensis* exposure. Following inoculation, ‘K10-1′ showed increased relative abundance in 42 metabolites and decreased abundance in 91 metabolites. In comparison, ‘K10-9′ registered increases in 50 metabolites and decreases in 61 metabolites (Figure 5a). Of these, 18 metabolites were commonly elevated in both cultivars post-inoculation, while 30 were consistently diminished (Figure 5b). KEGG enrichment analysis indicated that metabolites jointly augmented in both cultivars were mainly involved in ABC transporters (ko02010), nicotinate and nicotinamide metabolism (ko00760), and galactose metabolism (ko00052). In contrast, jointly reduced metabolites were primarily associated with flavonoid biosynthesis (ko00941), stilbenoid, diarylheptanoid, and gingerol biosynthesis (ko00945), as well as 2-oxocarboxylic acid metabolism (ko01210) (Appendix A).

Post-*P. cubensis* exposure, ‘K10-1′ displayed significant elevations in 24 metabolites, of which flavonoids constituted 33.3%, and reductions in 61 metabolites, with flavonoids and phenolic acids contributing to 22.9% and 19.6%, respectively (Figure 5b; Appendix A). In ‘K10-9′, 32 metabolites exhibited notably increased relative contents, with flavonoids comprising 12.5%, while 31 metabolites registered notable declines, with flavonoids and phenolic acids accounting for 32.3% and 19.4%, respectively (Figure 5b; Appendix A). In terms of specific flavonoid compounds post-infection, ‘K10-1′ experienced significant increases in eight and decreases in fourteen (Figure 5c). Meanwhile, ‘K10-9′ observed significant elevations in four and reductions in ten flavonoid compounds (Figure 5d). Collectively, these findings underscore pronounced alterations in the flavonoid profiles of both cultivars subsequent to *P. cubensis* inoculation.

### 2.6. Flavonoids’ Contribution to Melon’s Resistance against DM

To delve deeper into the contribution of the flavonoid biosynthesis pathway in melon’s defense against *P. cubensis* infection in ‘K10-1′ and ‘K10-9′ cultivars, we pinpointed 10 differentially expressed genes (DEGs) that play a role in this pathway (Appendix A). Post-infection with *P. cubensis*, the expression levels of several pivotal genes, including one chalcone synthase (CHS) gene (MELO3C014767.2), two flavonol synthase (FLS) genes (MELO3C018422.2 and MELO3C032760.2), two chalcone isomerase (CHI) genes (MELO3C020146.2 and MELO3C025484.2), and one naringenin 3-dioxygenase (F3H) gene (MELO3C035229.2), were observed to be markedly suppressed in both cultivars. Additionally, from the eleven differentially abundant metabolites (DAMs) associated with the flavonoid biosynthesis pathway post-infection, six DAMs were diminished in ‘K10-1′, while seven were curtailed in ‘K10-9′ (Appendix A). Each flavonoid metabolite had a significant correlation (*p*-value < 0.05), with the expression of at least one flavonoid biosynthesis-related gene (Appendix A). It is of particular significance that the concentrations of naringenin and pinobanksin, two crucial metabolites present in both cultivars, witnessed a substantial decline and showed significant correlation with eight of the ten flavonoid biosynthesis-related genes. Complementing these findings, a comprehensive evaluation of the total flavonoid content unveiled a significant reduction in both the ‘K10-1′ and ‘K10-9′ cultivars subsequent to *P. cubensis* inoculation (Figure 6a).

### 2.7. Activation of Lignin Biosynthesis Pathways in Both ‘K10-1′ and ‘K10-9′ Cultivars Post-P. cubensis Infection

The pathways of lignin and flavonoid biosynthesis serve as the two principal branches of phenylalanine metabolism [27]. Prior research indicates that alterations in flavonoid concentrations can influence lignin synthesis [28]. Our analyses discerned a marked decrease in the levels of 12 phenolic acid compounds, including lignin precursors, such as caffeic acid and coumaric acid, in ‘K10-1′ post-*P. cubensis* inoculation (Appendix A). This observation led us to postulate that the decline in lignin precursor and overall flavonoid content could be attributed to enhanced lignin biosynthesis. To validate this supposition, we examined both the gene expression profiles and metabolite levels pertinent to lignin formation. Using the acetyl bromide assay, we identified a significant elevation in the total lignin content in both ‘K10-1′ and ‘K10-9′ cultivars subsequent to *P. cubensis* exposure (Figure 6b). Additionally, post-inoculation, twenty-three lignin biosynthesis genes, including four phenylalanine ammonia-lyase (PAL), one trans-cinnamate 4-monooxygenase (C4H), and four 4-coumarate-CoA ligases (4CL), exhibited pronounced down-regulation in both cultivars (Figure 7a; Appendix A). Concurrently, key metabolites involved in the lignin biosynthesis pathway, namely phenylalanine, trans-5-O-caffeoyl shikimic acid, and 5-O-caffeoyl shikimic acid, manifested significant reductions in both ‘K10-1′ and ‘K10-9′. The correlation analysis results showed that 41 transcripts in the lignin biosynthetic pathway were correlated with 12 metabolites (*p*-value < 0.05) (Appendix A).

Upon quantifying lignin monomer content, we observed a prominent escalation in G/H/S monomer concentrations in both cultivars following *P. cubensis* inoculation. Among these, the H unit registered the maximum levels, succeeded by the G unit, while the S unit maintained the least presence (Figure 7b). Importantly, the H unit concentration in ‘K10-1′ consistently surpassed ‘K10-9′ in both pre- and post-*P. cubensis* exposure (Figure 7b). Collectively, these insights substantiate that in response to *P. cubensis* infection, both the resistant and susceptible cultivars shift their metabolic emphasis from flavonoid to lignin biosynthesis.

## 3. Discussion

Elucidating the mechanisms underlying the resistance of melons to *P. cubensis* is critical for refining integrated management strategies against DM. The objective of this study was to delineate the differential transcriptomic and metabolomic responses between the DM-resistant ‘K10-1′ variety and the DM-susceptible ‘K10-9′ variety upon *P. cubensis* infection. Although a prior study performed a transcriptomic analysis on cucumber leaves interacting with *P. cubensis*, thereby identifying a potential defensive pathway against DM [29], comprehensive knowledge about the melon leaf transcriptome and its regulatory role in conferring resistance to *P. cubensis* remains limited. Consequently, our research fills this knowledge gap by offering in-depth insights into the transcriptomic and metabolomic reactions of melons when infected with *P. cubensis*.

### 3.1. Different Basal Transcriptomes in Different Melon-Resistant Cultivars

One of the salient findings of this research was the distinct variation observed in the basal leaf transcriptomes between the ‘K10-1′ and ‘K10-9′ cultivars (Figure 2). A prevalent oversight in studies is the comparison of basal gene expression in hosts with varying disease susceptibilities before exposure to pathogens. Such baseline data are invaluable, as intrinsic differences in gene expression can offer clues to resistance or susceptibility patterns, potentially preconditioning the plant against diseases [30]. A study on two olive cultivars emphasized this by revealing differential root basal transcriptomes between a tolerant cultivar (Frantoio) and a susceptible one (Picual) when exposed to *Verticillium dahliae*, underscoring their essential defensive roles [20]. Our research pinpointed 1551 differentially expressed genes (DEGs) in uninfected leaves, wherein 651 DEGs were more expressive in ‘K10-1′ and 900 in ‘K10-9′ (Figure 2b). Post-infection with *P. cubensis*, 293 of the 651 genes in ‘K10-1′ still exhibited greater expression compared to those in ‘K10-9′ (Figure 2c and Appendix A). GO-enriched analysis results (q-value ≤ 0.05) indicated that DEGs with increased expression were prominently involved in fungal defense responses, symbiont interaction, and jasmonic acid hydrolase activities (Figure 2d). On the other hand, DEGs that were more expressive in ‘K10-9′ were predominantly associated with endopeptidase inhibitor activity, peptidase inhibitor activity, and phospholipid binding (Figure 2d).

Of the genes affiliated with fungal defense, an *FMO* gene exhibited notably higher expression in ‘K10-1′. Research has underscored the overexpression of *FMO1* in *Arabidopsis* to bolster resistance against *Pseudomonas* and *Hyaloperonospora parasitica* [31]. Several studies have underscored *FMO1*′s pivotal role in synthesizing N-hydroxypipecolic acid (NHP), which is recognized to amplify systemic acquired resistance (SAR) in *Arabidopsis* [22,32]. Another gene of interest was *FER*, encoding a receptor-like protein kinase. It modulates immunity by curbing the jasmonic acid (JA) and coronatine (COR) signals [23]. This gene manifested enhanced expression in ‘K10-1′ and was up-regulated in both cultivars upon exposure to *P. cubensis*.

Recent findings by Davoudi et al. spotlighted the elevated expression of HD-ZIP I/II and zf-HD, members of the homeobox transcription factor family, under drought stress [33]. Of these, *AtHB7* has been identified as instrumental in drought response based on *Arabidopsis* experiments [24,34]. In our work, *CmHB-7* exhibited a more pronounced expression in ‘K10-9′. Moreover, ‘K10-9′ showcased superior drought resilience compared to ‘K10-1′ (Appendix A). This suggests that each cultivar’s resistance or susceptibility to *P. cubensis* may be deeply anchored to their innate transcriptomic behaviors. A future avenue of interest for our research will be to investigate these fungus defense-associated genes as potential levers to elevate melon’s resilience against DM. Additionally, it has been documented that the *Oryza sativa* HD-Zip transcription factor *OsTFIL* directly binds to the promoters of lignin biosynthesis-related genes to promote the accumulation of lignin in shoots [35]. Our findings demonstrate a significant rise in lignin content in both melon cultivars following *P. cubensis* inoculation (Figure 6c). Consequently, there is a need for further investigation into the connection between *CmHB-7* and the lignin content in melon.

### 3.2. Response of Melon to DM Infection: Inhibiting Flavonoid Biosynthesis and Augmentation of Lignin Synthesis

The phenylpropanoid metabolic pathway, which comprises flavonoid and lignin branches, is integral to biotic stress resistance [27]. While many studies have posited the protective role of flavonoids against oomycete infections [36,37], our investigation revealed an unexpected decrease in total flavonoid content in both ‘K10-1′ and ‘K10-9′ cultivars post-inoculation with *P. cubensis* (Figure 6a). Remarkably, post-infection flavonoid concentrations in ‘K10-9′ consistently surpassed those in ‘K10-1′ (Figure 6a). Previous research has suggested that specific flavonoids might attract microorganisms toward plant colonization. For example, elevated phloridzin concentrations in apple leaves and shoots have been associated with increased vulnerability to *Valsa* canker due to the fungus *Valsa mali*, particularly in plants with *UGT88F1* gene overexpression [38]. 

In line with this, our metabolomic data revealed significant changes in several flavonoids in both cultivars following *P. cubensis* exposure, with a notable up-regulation of compounds, like quercetin and kaempferol, in ‘K10-1′ post-infection (Appendix A). The existing literature emphasizes the inhibitory role of glycosylated versions of these flavonoids in the growth of certain pathogens [9,39,40]. Moreover, while some studies indicate luteolin’s potential in inhibiting fungal spore germination [41], we observed its decline in ‘K10-9′ post-infection, a trend not discerned in ‘K10-1′. Consistent with this, there was a discernible suppression of multiple genes associated with flavonoid biosynthesis in both cultivars after *P. cubensis* exposure (Figure 5c,d). 

Recent research in cotton demonstrated a link between the suppression of lignin biosynthesis and enhanced resistance to pathogens [28]. Similarly, in tobacco, reduced flavonoid accumulation correlated with augmented lignin biosynthesis [42]. In our study, post-*P. cubensis* inoculation exhibited a reciprocal relationship between a decrease in flavonoids and an increase in lignin precursors and monomers, reiterating lignin’s established role as a defense modulator against pathogen invasion [43,44,45]. Furthermore, our data indicate a considerable increase in the content of lignin monomers G, H, and S in both cultivars post-inoculation, with the H monomer content being notably higher in ‘K10-1′ (Figure 7b).

The observed responses to *P. cubensis* infection suggest a strategic down-regulation in flavonoid biosynthesis genes, resulting in reduced flavonoid content and enhanced lignin synthesis in melon. The prominence of H-unit accumulation in ‘K10-1′ might further elucidate its superior resistance against this pathogen. A plethora of studies emphasize the responsive nature of lignin biosynthesis genes to pathogen attacks [20,21,46]. The enzymatic action of *HCT* highlights its vital role in diverting the metabolic flux toward either flavonoids or lignin production [27]. However, our findings, which showcased the suppression of *HCT* genes paired with decreased flavonoid and increased lignin content post-*P. cubensis* infection, signal a complex relationship that warrants more detailed investigations. Further research must prioritize validating these observations through targeted gene editing or overexpression strategies targeting the *HCT* gene, facilitating a more comprehensive elucidation of the roles attributed to lignin and flavonoids in melon’s defense against *P. cubensis*.

## 4. Materials and Methods

### 4.1. Plant Material and Treatment

The melon (*Cucumis melo* L.) genotypes, specifically the DM-resistant resource ‘PI 442177′ (ID: ‘K10-1′) and the susceptible landrace ‘Huangtu’ (ID: ‘K10-9′), are preserved at the Hami Melon Research Center under the aegis of the Xinjiang Academy of Agricultural Sciences, Xinjiang, China. In winter 2022, an isolate of *P. cubensis* was sourced from melon leaves exhibiting symptoms of a natural infection. This specimen was procured from the Hainan Sanya Crop Breeding Experiment Center, affiliated with the Xinjiang Academy of Agricultural Sciences, Hainan, China.

Experiments were conducted to assess the responses of ‘K10-1′ and ‘K10-9′ to *P. cubensis* inoculation at both seedling and mature stages, following the artificial inoculation procedure described by Zhang et al. [2]. Each experiment incorporated a minimum of 3 biological replicates, with 30 plants from each cultivar per replicate. Once the first true leaf fully expanded, seedlings were inoculated with *P. cubensis* using a spray method, in accordance with Zhang et al. [2]. The spore concentration for inoculation was set at 5 × 10^3^ spores/mL or higher, supplemented with 0.5 ‰ Tween, while double-distilled water (ddH_2_O) served as the control. Leaf samples from both the treated and control groups were collected at 0 and 48 h post-inoculation (hpi), immediately flash-frozen in liquid nitrogen, and stored at −80 °C for subsequent transcriptomic and metabolomic analyses. The phenotypic responses were documented at intervals of 3, 5, 7, and 10 days post-inoculation (dpi). During the mature stage, DM occurrence was recorded at 10 and 15 dpi. Disease severity was quantified on a scale ranging from 0 to 5, where 0 represented the utmost resistance and 5 denoted extreme susceptibility, as detailed by Zhang et al. [2].

The drought stress tolerance of ‘K10-1′ and ‘K10-9′ was assessed under controlled illumination conditions in growth chambers. Upon reaching the three-leaf stage, the plants designated for the experimental groups were exposed to a Hoagland nutrient solution supplemented with 20% PEG-4000 to mimic drought conditions. Meanwhile, the control group received only the standard Hoagland nutrient solution. Phenotypic observations of the plants were documented at intervals of 0, 2, 12, 18, 24, and 48 h following the initiation of the drought treatment.

### 4.2. RNA Sample Preparation and Transcriptome Sequencing

The RNA sample preparation and subsequent transcriptomic sequencing were conducted by MetWare Company (Metware Biotechnology Co., Ltd., Wuhan, China). Sequencing libraries were prepared using the NEBNext^®^ Ultra^TM^ RNA Library Prep Kit for Illumina^®^ (New England Biolabs, Ipswich, MA, USA), following the manufacturer’s instructions. The AMPure XP system (Beckman Coulter, Beverly, MA, USA) was employed for library fragment purification. Once the quality of the library was ascertained using the Agilent Bioanalyzer 2100 system, the sequencing process was undertaken on the Illumina HiSeqTM 2500 platform (Illumina, CA, USA). The high-quality reads, post-filtration, were aligned to the reference genome (DHL92_v3.6.1) [47] using HISAT (Version 2.1.0) [48].

### 4.3. Gene Expression Quantification and Differential Analysis

For gene alignment, featureCounts (Version 1.6.2) [49] was employed, whereas StringTie (Version1.3.4d) [48] was utilized for FPKM calculations. The differential expression analysis between the two groups was undertaken using DESeq2 v1.22.1 [50]. *p*-values were adjusted employing the Benjamini and Hochberg method. Differentially expressed genes (DEGs) were identified based on the following criteria: a *p*-value (FDR) of <0.05 and |log_2_ Fold Change| ≥ 1. The gene ontology (GO) and KEGG enrichment analysis of the DEGs was conducted using the clusterProfiler R package [51], with a significance threshold set at an adjusted *p*-value of <0.05.

### 4.4. Metabolite Extraction and UPLC-ESI-MS/MS Analysis

Metabolomic evaluations were carried out by the MetWare Company (Metware Biotechnology Co., Ltd., Wuhan, China). Melon leaf samples were subjected to freeze drying using a Scientz-100F vacuum freeze dryer. The lyophilized samples were subsequently pulverized using an MM 400 mixer mill (Retsch). For the extraction process, a quantity of 50 mg of the powdered sample was solubilized in 1.2 mL of 70% methanol, and this mixture underwent six intermittent vortexing sessions, each lasting for 30 s within a 30 min interval. Following centrifugation at 12,000 rpm for 3 min, the supernatants were sieved through a 0.22 μm pore size filter (SCAA-104, 0.22 μm pore size; ANPEL, Shanghai, China). The filtrates were then subjected to Ultra-Performance Liquid Chromatography (UPLC) using a SHIMADZU Nexera X2 system, paired with Tandem Mass Spectrometry (MS/MS) on an Applied Biosystems 6500 QTRAP. The resultant mass spectrometry data were processed using Analyst (version 1.6.3). The MetWare database (MWDB) was utilized for both qualitative and quantitative assessment of the sample metabolites via mass spectrometry.

### 4.5. Data Processing and Metabolite Identification

For further analysis, metabolites possessing a VIP (Variable Importance in Projection) value ≥ 1 from the OPLS-DA were chosen. The differential accumulation of metabolites between comparative groups was discerned using a Fold Change criterion: metabolites demonstrating a Fold Change ≥ 2 or ≤ 0.5 were considered significant. Principal Component Analysis (PCA) was conducted by employing the prcomp function in the R statistical software (www.r-project.org (accessed on 31 October 2023)). The identified metabolites underwent annotation using the KEGG Compound database (http://www.kegg.jp/kegg/compound/ (accessed on 8 September 2023)). These annotated metabolites were subsequently aligned to the KEGG Pathway database (http://www.kegg.jp/kegg/pathway.html (accessed on 30 October 2023)). Pathways associated with these significantly regulated metabolites underwent a Metabolite Sets Enrichment Analysis (MSEA). The significance of the identified pathways was established using a *p*-value derived from the hypergeometric test.

### 4.6. Quantitative Real-Time Polymerase Chain Reaction (qRT-PCR) Analysis

Total RNA was extracted from 100 mg of leaf tissue derived from both ‘K10-1′ and ‘K10-9′ plants, which were either inoculated with *P. cubensis* or maintained as untreated controls. The extraction process was conducted using a TaKaRa MiniBEST Plant RNA Extraction Kit (Takara Biotechnology Co., Ltd., Dalian, China). Subsequently, the extracted RNA underwent reverse transcription into cDNA with a PrimeScript™ 1st Strand cDNA Synthesis Kit (Takara). For normalization, the internal control employed was the melon ADP ribosylation factor 1 gene (GenBank ID: MU47713, CmADP). The PCR mixture, with a total volume of 20 μL, comprised 10 μL of 2×SuperReal PreMix Plus, 10 ng of template cDNA, and 0.6 μL of both the forward and reverse primers, each at a concentration of 10 μM.

qRT-PCR assays were conducted on a Roche Light-Cycler480R PCR system. The amplification conditions included an initial polymerase activation at 95 °C for 15 min, followed by 40 cycles: denaturation at 95 °C for 10 s, annealing at 60 °C for 20 s, and extension at 72 °C for 20 s. The relative gene expression was quantified using the 2^−ΔΔCt^ method, referencing the Ct value determined at the specific fluorescence threshold for each sample [52]. The expression of each gene was assessed through three distinct PCR replicates, with the mean value representing the gene’s expression level. The primers employed in the qRT-PCR assays are delineated in Appendix A.

### 4.7. Determination of Total Flavonoid Content

An AlCl_3_ colorimetric assay was employed to quantify the total flavonoid content, with modifications from the method delineated by Shraim et al. [53]. Specifically, leaves from both mock-treated and *P. cubensis*-inoculated ‘K10-1′ and ‘K10-9′ plants were collected at 48 hpi and subsequently ground in the presence of liquid nitrogen. From the resulting powder, 100 mg was suspended in 2 mL of 80% methanol. After centrifugation at 13,000 rpm, the supernatant was treated with 30 µL of 5% NaNO_2_ (*w*/*v*) and allowed to stand at room temperature for 5 min. This was followed by the addition of 60 µL of 10% AlCl3 (*w*/*v*), with a subsequent 5 min incubation. To this mixture, 0.2 mL of 1 M NaOH was added. Absorbance was measured at 500 nm using a NanoDrop ND-2000 spectrophotometer (Thermo Scientific, Wilmington, DE, USA). A rutin standard was utilized for calibration.

### 4.8. Determination of Total Lignin Content and Lignin Composition

The lignin content in the leaf tissue was determined using the acetyl bromide method, adapted from the protocol presented by Barnes and Anderson [54]. Leaf samples, each weighing approximately 100 mg (with precise weights recorded), were finely ground in the presence of liquid nitrogen. These samples were subsequently subjected to sequential washes with ethanol, 95% ethanol, and deionized water to eliminate soluble substances. After freeze drying, each sample was treated with 5 mL of an acetyl bromide–glacial acetic acid solution (1:3, *v*/*v*) and incubated at 70 °C for 30 min. The reaction mixture was then neutralized with the addition of 0.9 mL of 2 M NaOH and 3 mL of glacial acetic acid. To terminate the reaction, 1 mL of a 7.5 M hydroxylamine hydrochloride solution was introduced. Following centrifugation at 13,000 rpm, the supernatant’s absorbance was assessed at 280 nm using a NanoDrop ND-2000 spectrophotometer (Thermo Scientific). Alkali lignin was used as the calibration standard. For the determination of lignin monomer composition within the leaves, thioacidolysis followed by gas chromatography–mass spectrometry analysis was performed on three biological replicates, in accordance with the method described by Chen et al. [55]. The lignin monomer content was expressed as a percentage of the dry weight.

## 5. Conclusions

In the present study, we conducted a detailed comparison of transcriptomic and metabolomic profiles between the *P. cubensis*-resistant cultivar ‘K10-1′ and its susceptible counterpart ‘K10-9′, both prior to and following *P. cubensis* inoculation. Utilizing KEGG and GO analyses, we identified multiple pathways and genes integral to *P. cubensis* resistance. Intriguingly, our results indicate that the inherent basal resistance of ‘K10-1′ is a major determinant of its resilience to *P. cubensis*. Moreover, both the resistant and susceptible cultivars seem to respond to *P. cubensis* infection by reducing flavonoid biosynthesis and increasing lignin production. These findings provide a foundational understanding, paving the way for future genetic studies to further investigate the contributions of lignin and basal resistance to melon’s defense strategies against *P. cubensis*.

## Figures and Tables

**Figure 1 ijms-24-17552-f001:**
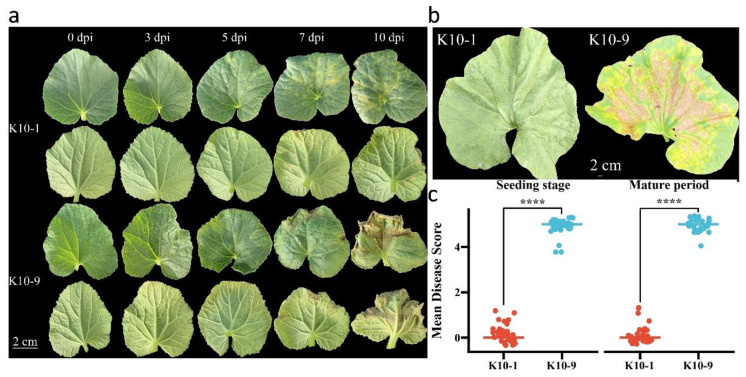
Phenotypic responses of melon to *P. cubensis* infection. (**a**) Phenotypic observations of seedling leaves ‘K10-1′ and ‘K10-9′ were recorded at intervals of 3, 5, 7, and 10 days post-inoculation (dpi). (**b**) Field phenotypes of leaves from ‘K10-1′ and ‘K10-9′ were documented at 15 dpi. (**c**) Dispersed boxplots present the DM scores for ‘K10-1′ and ‘K10-9′ across various stages (*n* = 30 for ‘K10-1′ and ‘K10-9′). Both seedlings and field assays were executed with a minimum of three replicates, consistently yielding similar results. Each replication comprised at least 30 plants. A *t*-test revealed a statistically significant difference (**** *p* < 0.0001) between ‘K10-1′ and ‘K10-9′.

**Figure 2 ijms-24-17552-f002:**
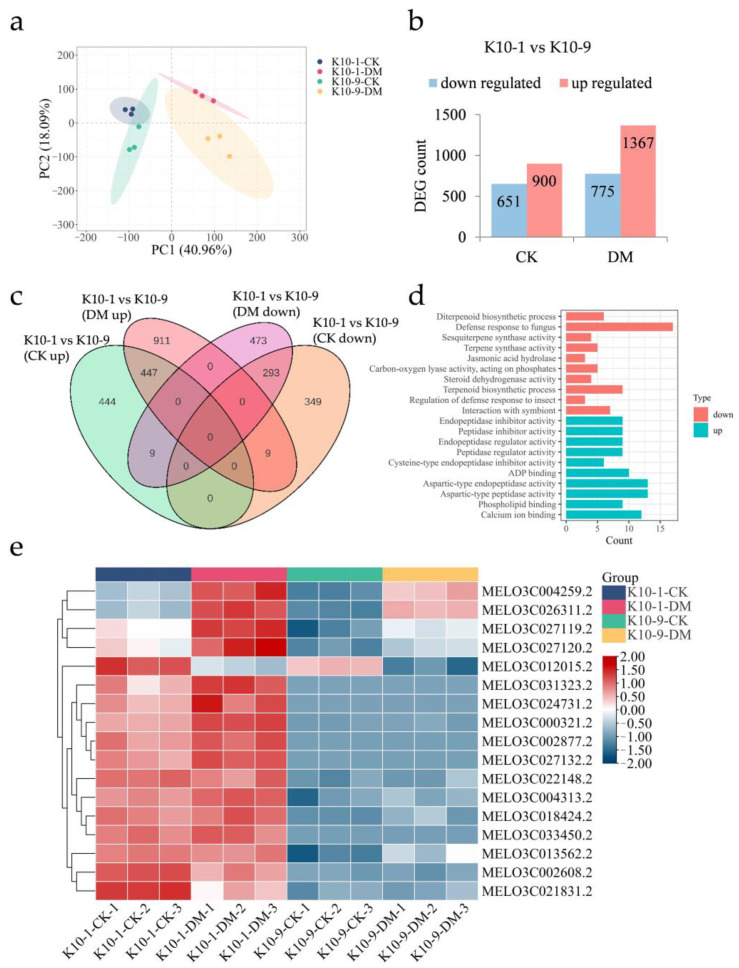
Transcriptomic analysis of resistant ‘K10-1′ and susceptible ‘K10-9′’ leaves following *P. cubensis* infection. (**a**) PCA of the transcriptome data differentiates samples: ‘CK’ represents non-manipulated controls, ‘DM’ signifies samples collected 48 h post-P. *cubensis* inoculation. (**b**) Tabulation of differentially expressed genes (DEGs) showcases both up-regulated and down-regulated genes when contrasting ‘K10-1′ and ‘K10-9′ under both treated and control conditions. Here, ‘K10-1′ serves as the resistance reference. (**c**) A Venn diagram delineates the DEGs that are both unique and shared between the resistant ‘K10-1′ and the susceptible ‘K10-9′ across both treatment and control groups. (**d**) Gene ontology (GO) enrichment analysis identifies terms related to the 293 DEGs consistently expressed at higher levels in ‘K10-1′ compared to ‘K10-9′ and the 447 DEGs consistently expressed at lower levels under both treatment and control conditions. A threshold q-value of ≤ 0.05 was maintained. (**e**) A heat map visually represents the expression patterns of 17 genes associated with fungal defense responses, which consistently exhibit higher expression levels in ‘K10-1′ than in ‘K10-9′ across both the control and treatment groups in melon leaves.

**Figure 3 ijms-24-17552-f003:**
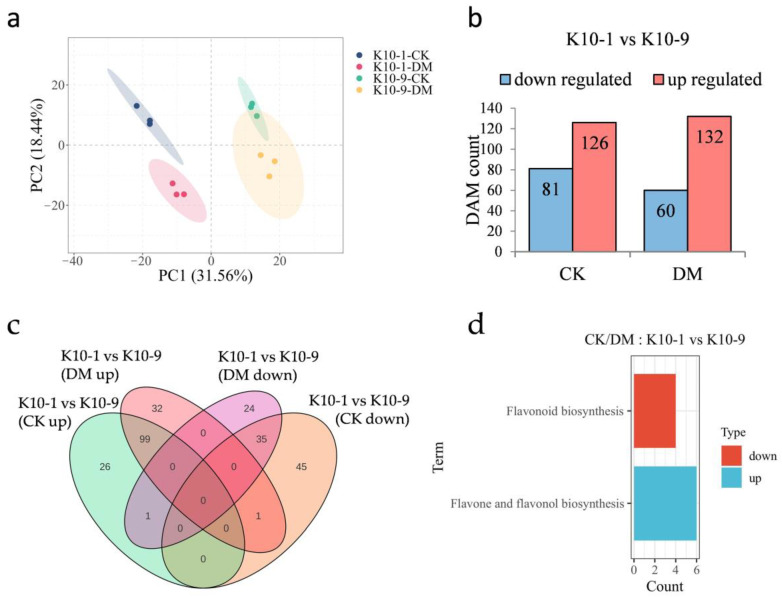
Metabolomic profiling of resistant ‘K10-1′ and susceptible ‘K10-9′ leaves following *P. cubensis* infection. (**a**) PCA delineates variations within the metabolomic dataset. (**b**) A comprehensive list presents both up-regulated and down-regulated differentially accumulated metabolites (DAMs) when contrasting ‘K10-1′ and ‘K10-9′ under treated and control conditions. In this context, ‘K10-1′ functions as the reference for resistance. (**c**) A Venn diagram differentiates the DAMs that are either unique or shared between the resistant ‘K10-1′ and the susceptible ‘K10-9′ across both the treatment and control conditions. (**d**) KEGG enrichment analysis reveals pathways related to the up-regulated and down-regulated DAMs identified between ‘K10-1′ and ‘K10-9′ under both treated and control scenarios. A significance threshold q-value of ≤0.05 was maintained.

**Figure 4 ijms-24-17552-f004:**
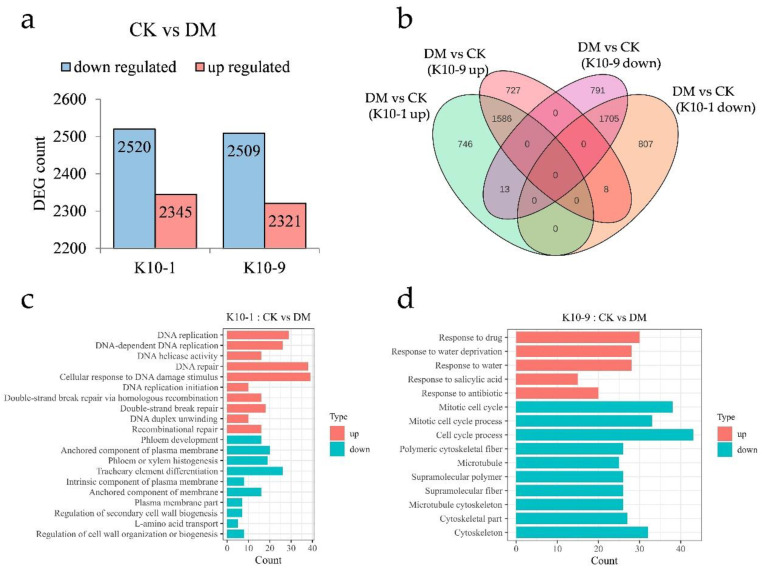
Transcription dynamics in resistant ‘K10-1′ and susceptible ‘K10-9′ leaves following *P. cubensis* infection. (**a**) An enumeration of differentially expressed genes (DEGs) reveals both up-regulated and down-regulated genes when contrasting treatment and control conditions in both ‘K10-1′ and ‘K10-9′. (**b**) A Venn diagram illustrates the DEGs that are distinct or shared between the treatment and control groups in both ‘K10-1′ and ‘K10-9′. (**c**) Gene ontology (GO) enrichment analysis highlights pathways associated with up-regulated and down-regulated DEGs, specifically in ‘K10-1′. (**d**) GO enrichment pathways for DEGs, either up-regulated or down-regulated, are delineated exclusively for ‘K10-9′. A significance threshold q-value of ≤0.05 was adhered to.

**Figure 5 ijms-24-17552-f005:**
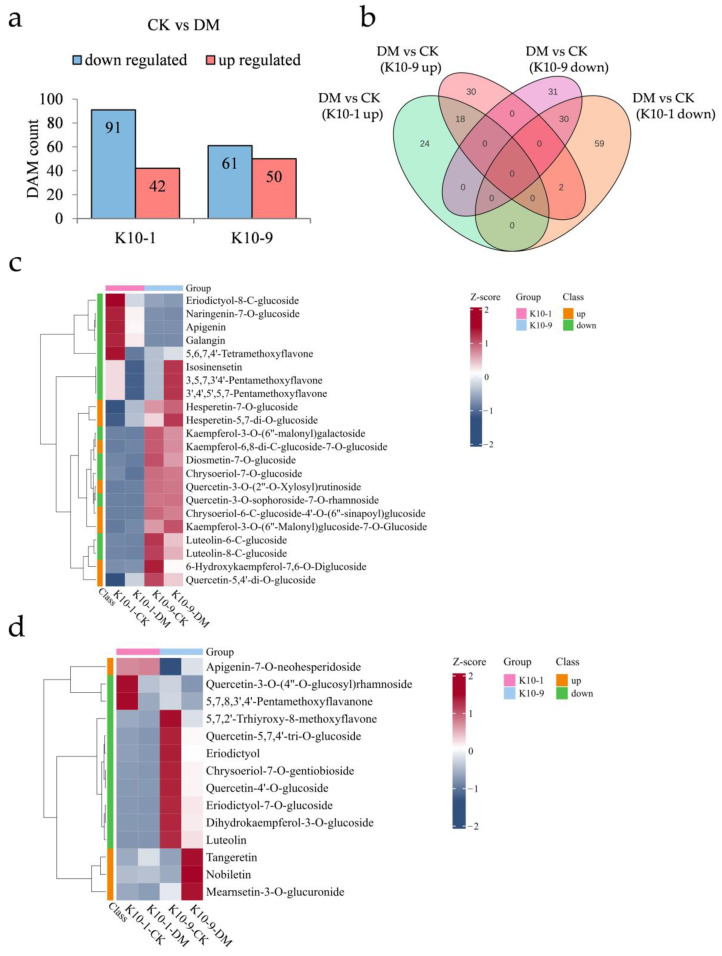
Metabolic shifts in resistant ‘K10-1′ and susceptible ‘K10-9′ leaves following *P. cubensis* infection. (**a**) Analysis reveals a comprehensive count of differentially accumulated metabolites (DAMs) that were either up-regulated or down-regulated when contrasting the treatment with the control conditions for both ‘K10-1′ and ‘K10-9′. (**b**) A Venn diagram elucidates DAMs that are either distinct or shared between the treatment and control scenarios in the cultivars ‘K10-1′ and ‘K10-9′. (**c**) A heat map underscores the flavonoids that exhibited either heightened or diminished accumulation exclusively in ‘K10-1′ when comparing the treatment and control conditions. (**d**) A heat map for ‘K10-9′ portrays the accumulation pattern of up-regulated and down-regulated flavonoids in melon leaves when the treatment is juxtaposed with the control.

**Figure 6 ijms-24-17552-f006:**
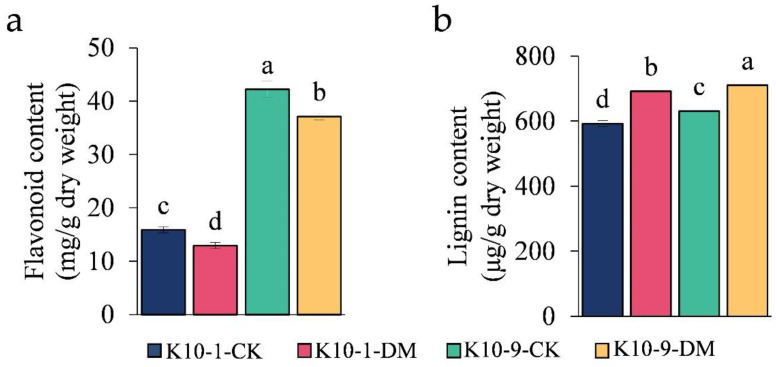
Analysis of content variability in (**a**) total flavonoids; (**b**) total lignin. Each error bar denotes the mean ± standard deviation derived from three independent biological replicates. Distinct letters signify statistically significant differences (*p* < 0.05) among groups, as determined by Duncan’s multiple range test.

**Figure 7 ijms-24-17552-f007:**
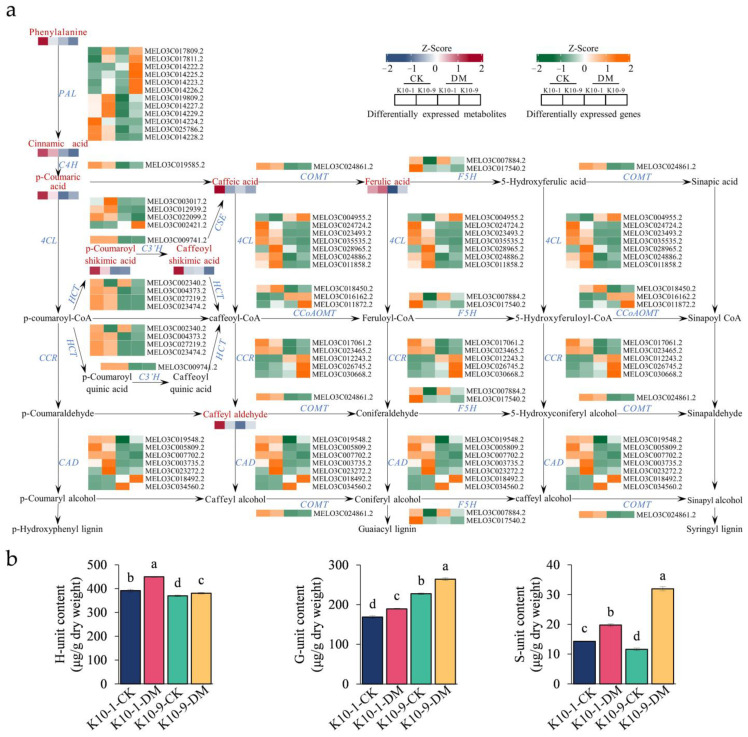
Expression and accumulation profiles of lignin biosynthesis-related genes and metabolites in resistant ‘K10-1′ and susceptible ‘K10-9′ pre- and post-inoculation with *P. cubensis*. (**a**) Profiling of transcripts and metabolites related to the 47 differentially expressed genes (DEGs) and 12 differentially accumulated metabolites (DAMs) implicated in lignin biosynthesis pathways in both ‘K10-1′ and ‘K10-9′. These include PAL (phenylalanine ammonia-lyase), C4H (trans-cinnamate 4-monooxygenase), 4CL (4-coumarate-CoA ligase), CCR (cinnamoyl-CoA reductase), CAD (cinnamyl-alcohol dehydrogenase), HCT (shikimate O-hydroxycinnamoyltransferase), C3′H (5-O-(4-coumaroyl)-D-quinate 3′-monooxygenase), CCoAOMT (caffeoyl-CoA O-methyltransferase), COMT (caffeic acid 3-O-methyltransferase), F5H (ferulate-5-hydroxylase), and CSE (caffeoylshikimate esterase). (**b**) Analysis of G, H, and S monomers present in the leaves of ‘K10-1′ and ‘K10-9′ across both the treatment and control groups. Each error bar illustrates the mean ± standard deviation derived from three independent biological replicates. Distinct letters denote statistically significant differences (*p* < 0.05) among groups, as determined by Duncan’s multiple range test.

## Data Availability

Data are contained within the article and Appendix A.

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
