# Peer review of "Comparative Analysis of Transcriptomics and Metabolomics Reveals Defense Mechanisms in Melon Cultivars against Pseudoperonospora cubensis Infection"

_ijms, 2023, doi:10.3390/ijms242417552_

Round 1

Reviewer 1 Report

Comments and Suggestions for Authors

Dear Authors,

I have read the submitted manuscript titled „Comparative Analysis of Transcriptomics and Metabolomics Reveals Defense Mechanisms in Melon Cultivars Against Pseudoperonospora cubensis Infection” with considerable interest. I think, that it contains important results, which might interest the international audience. Nevertheless, I have found some flawns, which (in my opinion) should be improved or at least clarified before an eventual publication. I have listed them below:

1.       I suggest to extent the bacground of investigations i.e. slighty enlarge the botanical description of Cucumis melo L. and point out its nutritional values. Moreover, the description of Pseudoperonospora cubensis should be improved.

2.       Please, improve the quality of Figures, some fragments are illegible.

3.       Please, look into the following publications. Perhaps, some of the would be help ful in manuscript improvements:

·         T. Kerje, M. Grum 2000. The origin of melon, Cucumis melo: a review of the literature. Acta Hortic. 510, 37-44. DOI: 10.17660/ActaHortic.2000.510.5

·         James D. Mccreight, Haim Nerson, Rebecca Grumet. 1993. 20 - Melon: Cucumis melo L., Editor(s): G. Kalloo, B.O. Bergh, Genetic Improvement of Vegetable Crops, Pergamon, Pages 267-294.

·         Lester, G. 1997. Melon (Cucumis melo L.) Fruit Nutritional Quality and Health Functionality. HortTechnology horttech, 7(3), 222-227.

·         Manchali, Shivapriya, Kotamballi N. Chidambara Murthy, Vishnuvardana, and Bhimanagouda S. Patil. 2021. "Nutritional Composition and Health Benefits of Various Botanical Types of Melon (Cucumis melo L.)" Plants 10, no. 9: 1755. https://doi.org/10.3390/plants10091755

·         Lebeda, A., Cohen, Y. Cucurbit downy mildew (Pseudoperonospora cubensis)—biology, ecology, epidemiology, host-pathogen interaction and control. Eur J Plant Pathol 129, 157–192 (2011). https://doi.org/10.1007/s10658-010-9658-1

Author Response

Dear Editorial Board of International Journal of Molecular Sciences,

We thank the reviewers for their comments. We are pleased to submit our revised manuscript entitled “Comparative Analysis of Transcriptomics and Metabolomics Reveals Defense Mechanisms in Melon Cultivars Against Pseudoperonospora cubensis Infection”, which we believe has been strengthened by our responses to the reviewer comments (see below point-to-point response).

We have extended the botanical description of Melon and pointed out its nutritional values. we have removed some older publications and added valuable publications recommended by the Reviewer. The quality of all pictures was improved to 300dpi and we also recommended research directions in future research.

We hope that you find these revisions have improved the manuscript sufficiently for publication. We look forward to receiving comments on our revised manuscript.

Sincerely,

Xuejun Zhang

Hami-melon Research Center, Xinjiang Academy of Agricultural Sciences, Urumqi 830091, China

Point-by-Point Response to Reviewers’ Comments

Reviewer #1

I have read the submitted manuscript titled “Comparative Analysis of Transcriptomics and Metabolomics Reveals Defense Mechanisms in Melon Cultivars Against Pseudoperonospora cubensis Infection” with considerable interest. I think, that it contains important results, which might interest the international audience. Nevertheless, I have found some flawns, which (in my opinion) should be improved or at least clarified before an eventual publication. I have listed them below:

Thanks for your comments!

  1. I suggest to extent the bacground of investigations i.e. slighty enlarge the botanical description of Cucumis melo L. and point out its nutritional values. Moreover, the description of Pseudoperonospora cubensis should be improved.

Thanks! We have added related descriptions in the Introduction sections on lines 41-56 in the revised manuscript.

  1. Please, improve the quality of Figures, some fragments are illegible.

Thanks for pointing this out! We have redrawn all the Figures in the revised manuscript and adjusted the clarity of each Figure to be greater than 300 dpi.

  1. Please, look into the following publications. Perhaps, some of the would be helpful in manuscript improvements:

Thank you to the reviewers for providing us with the following publications. We have carefully read and found all of these publications were very helpful for improved writing of our manuscript and will be valuable for future research. However, Reviewer 3 suggested removing publications older than 10 years. Therefore, we have deleted or replaced most references older than 10 years in the manuscript. Finally, we have cited one publication (Manchali, Shivapriya, Kotamballi N. Chidambara Murthy, Vishnuvardana, and Bhimanagouda S. Patil. 2021. "Nutritional Composition and Health Benefits of Various Botanical Types of Melon (Cucumis melo L.)" Plants 10, no. 9: 1755. https://doi.org/10.3390/plants10091755) on line 45 recommended by Reviewer 1.

Reviewer 2 Report

Comments and Suggestions for Authors

Reviewer’s Comments

“Comparative Analysis of Transcriptomics and Metabolomics Reveals Defense Mechanisms in Melon Cultivars Against Pseudoperonospora cubensis Infection” by Ling et al., is a very interesting and well written manuscript. It also outlines the possible mechanisms by which P. cubensis infection causes both resistant and susceptible cultivars of melons to suppress flavonoid biosynthesis via the downregulation of the associated genes, whilst concurrently promoting lignin production. I will recommend the publication of this manuscript upon the minor revision.

Major Modifications

1.     Line 19-20. “Downy mildew (DM), a pervasive foliar disease, poses a significant threat to global melon production. Although several quantitative trait loci related to DM resistance have been identified, the comprehensive genetic underpinnings of this tolerance remain largely uncharted.” This sentence does not correlate with the initial thought or idea of resistance. Please revise. E.g. the use of the term “tolerance” cannot be equated to “resistance”

2.     Line 22. “Unravel the molecular mechanisms strategies”, please revise this phrase for better comprehension and flow.

3.     Line 26. Replace “confronted” with a more appropriate terminology.  

4.     Line 41. The referencing format “(https://www.fao.org/home/zh)” as contained in this line is not in conformity with the rest of the work. E.g. see line 46 “[1]”. Please revise appropriately.

5.      Line 43. The referencing format “(Berk. & Curt.)” as contained in this line is not in conformity with the rest of the work. E.g. see line 46 “[1]”. Please revise appropriately.

6.     Line 52-55. Reference required in the statement “In recent years, there has been an increasing emphasis among researchers on leveraging multi-omics data, encompassing metabolomics and transcriptomics, to elucidate metabolic pathways and molecular regulatory networks integral to plant disease resistance mechanisms”

7.     Figure 2c is not well-labeled to enhance complete understanding of the figure. It is difficult to look at the figure and know which sector of the Venn diagram belonged to the resistant Melon and which belonged to the susceptible. E.g. See Figure 4b and 5b.

8.     Line 173. The phrase “heightened expressed” should be replaced with “heightened expression”

9.     Lines 175-179. “In contrast, the 447 genes persistently registered diminished expression in comparison ‘K10-9’ were predominantly enriched in processes such as endopeptidase inhibitor activity (GO:0004866), peptidase inhibitor activity (GO:0030414), and phospholipid binding (GO:0005543) (Figure 2d).” needs revision. Please revise appropriately.

10. Figure 3c is not well-labeled to enhance complete understanding of the figure. It is difficult to look at the figure and know which sector of the Venn diagram belonged to the resistant Melon and which belonged to the susceptible. E.g. See Figure 4b and 5b.

11. Line 209. As shown in Figure 3c this phrase “35 metabolites consistently registered elevated levels” seems incorrect. Please revise.

12. Line 210. As shown in Figure 3c this phrase “while 99 metabolites consistently lagged behind” seems incorrect. Please revise.

13. Line 213. As shown in Figure 3c this phrase “the 35 metabolites with heightened abundance” seems incorrect. Please revise.

14. Line 216. As shown in Figure 3c this phrase “the 99 metabolites with reduced abundance” seems incorrect. Please revise.

Author Response

Dear Editorial Board of International Journal of Molecular Sciences,

We thank the reviewers for their comments. We are pleased to submit our revised manuscript entitled “Comparative Analysis of Transcriptomics and Metabolomics Reveals Defense Mechanisms in Melon Cultivars Against Pseudoperonospora cubensis Infection”, which we believe has been strengthened by our responses to the reviewer comments (see below point-to-point response).

We have extended the botanical description of melon and pointed out its nutritional value. We have removed some older publications and added valuable publications recommended by the Reviewer. The quality of all pictures was improved to 300 dpi and we also recommended research directions in future research.

We hope that you find these revisions have improved the manuscript sufficiently for publication. We look forward to receiving comments on our revised manuscript.

Sincerely,

Xuejun Zhang

Hami-melon Research Center, Xinjiang Academy of Agricultural Sciences, Urumqi 830091, China

Point-by-Point Response to Reviewers’ Comments

Reviewer #2

Reviewer’s Comments

“Comparative Analysis of Transcriptomics and Metabolomics Reveals Defense Mechanisms in Melon Cultivars Against Pseudoperonospora cubensis Infection” by Ling et al., is a very interesting and well written manuscript. It also outlines the possible mechanisms by which P. cubensis infection causes both resistant and susceptible cultivars of melons to suppress flavonoid biosynthesis via the downregulation of the associated genes, whilst concurrently promoting lignin production. I will recommend the publication of this manuscript upon the minor revision.

Thanks for your comments!

Major Modifications

  1. Line 19-20. “Downy mildew (DM), a pervasive foliar disease, poses a significant threat to global melon production. Although several quantitative trait loci related to DM resistance have been identified, the comprehensive genetic underpinnings of this tolerance remain largely uncharted.” This sentence does not correlate with the initial thought or idea of resistance. Please revise. E.g. the use of the term “tolerance” cannot be equated to “resistance”

Thanks for pointing this out. We have changed “tolerance” to “resistance”.

  1. Line 22. “Unravel the molecular mechanisms strategies”, please revise this phrase for better comprehension and flow.

Thank you! We have revised this sentence in lines 24-25.

  1. Line 26. Replace “confronted” with a more appropriate terminology.

Thanks for pointing this out, we have changed “confronted with” to “infected” in lines 28.

  1. Line 41. The referencing format “(https://www.fao.org/home/zh)” as contained in this line is not in conformity with the rest of the work. E.g. see line 46 “[1]”. Please revise appropriately.

Thanks for the reviewer's reminder! We have revised the referencing format in line 48.

  1. Line 43. The referencing format “(Berk. & Curt.)” as contained in this line is not in conformity with the rest of the work. E.g. see line 46 “[1]”. Please revise appropriately.

Thanks for the reviewer's reminder! We have revised the referencing format in line 56.

  1. Line 52-55. Reference required in the statement “In recent years, there has been an increasing emphasis among researchers on leveraging multi-omics data, encompassing metabolomics and transcriptomics, to elucidate metabolic pathways and molecular regulatory networks integral to plant disease resistance mechanisms”

Thanks for pointing this out. We have added references in lines 69 in the revised manuscript.

  1. Figure 2c is not well-labeled to enhance complete understanding of the figure. It is difficult to look at the figure and know which sector of the Venn diagram belonged to the resistant Melon and which belonged to the susceptible. E.g. See Figure 4b and 5b.

We agree with the reviewer’s opinion. We have added the names of the plant material in Figures 2c, 3c, 4b, and 5b and added information about the disease resistance of the two materials in the figure legend in lines 751, 763, 772, and 780.

  1. Line 173. The phrase “heightened expressed” should be replaced with “heightened expression”

Done in line 168 of revised manuscript currently.

  1. Lines 175-179. “In contrast, the 447 genes persistently registered diminished expression in comparison ‘K10-9’ were predominantly enriched in processes such as endopeptidase inhibitor activity (GO:0004866), peptidase inhibitor activity (GO:0030414), and phospholipid binding (GO:0005543) (Figure 2d).” needs revision. Please revise appropriately.

We agree with the reviewer’s opinion and revised it in lines 171-176.

  1. Figure 3c is not well-labeled to enhance complete understanding of the figure. It is difficult to look at the figure and know which sector of the Venn diagram belonged to the resistant Melon and which belonged to the susceptible. E.g. See Figure 4b and 5b.

As mentioned above, we have added the names of the plant material in Figures 2c, 3c, 4b, and 5b and added information about the disease resistance of the two materials in the figure legend in lines 751, 763, 772, and 780.

  1. Line 209. As shown in Figure 3c this phrase “35 metabolites consistently registered elevated levels” seems incorrect. Please revise.

Thank you! We have rewritten this sentence in lines 201-204.

  1. Line 210. As shown in Figure 3c this phrase “while 99 metabolites consistently lagged behind” seems incorrect. Please revise.

Thank you! We have rewritten this sentence in lines 201-204.

  1. Line 213. As shown in Figure 3c this phrase “the 35 metabolites with heightened abundance” seems incorrect. Please revise.

Thank you! We have replaced abundance with content.

  1. Line 216. As shown in Figure 3c this phrase “the 99 metabolites with reduced abundance” seems incorrect. Please revise.

Thank you! We have replaced abundance with content.

Reviewer 3 Report

Comments and Suggestions for Authors

The manuscript contains interesting experimental results on the defence mechanisms of melon (Cucumis melo L.) against P. cubensis. The manuscript needs minor revisions.

Please detail the recommended research directions in the near future.

Comments

Line 41, please cite publications as recommended by the Editor.

Please provide details of the manufacturer of the statistical software used to analyse the study data.

References

Please remove publications older than 10 years.

Author Response

Dear Editorial Board of International Journal of Molecular Sciences,

We thank the reviewers for their comments. We are pleased to submit our revised manuscript entitled “Comparative Analysis of Transcriptomics and Metabolomics Reveals Defense Mechanisms in Melon Cultivars Against Pseudoperonospora cubensis Infection”, which we believe has been strengthened by our responses to the reviewer comments (see below point-to-point response).

We have extended the botanical description of melon and pointed out its nutritional values. we have removed some older publications and added valuable publications recommended by the Reviewer. The quality of all pictures was improved to 300 dpi and we also recommended research directions in future research.

We hope that you find these revisions have improved the manuscript sufficiently for publication. We look forward to receiving comments on our revised manuscript.

Sincerely,

Xuejun Zhang

Hami-melon Research Center, Xinjiang Academy of Agricultural Sciences, Urumqi 830091, China

Point-by-Point Response to Reviewers’ Comments

Reviewer #3

The manuscript contains interesting experimental results on the defence mechanisms of melon (Cucumis melo L.) against P. cubensis. The manuscript needs minor revisions.

Please detail the recommended research directions in the near future. ?

We sincerely appreciate this comment from reviewers. Our research has identified the significance of the basal transcriptome in melon resistance to powdery mildew. Therefore, a future research direction is to investigate whether these basal transcriptomes play an important role in melon against P. cubensis. We have also discovered the involvement of lignin in melon against P. cubensis, and CmHB7, as a member of the HD-ZIP transcription factor family, poses a potential avenue for future research on whether it regulates lignin to contribute to melon resistance against powdery mildew. We have included these prospects in lines 371-373 and lines 377-379 of the discussion section. Furthermore, HCT plays a vital role in diverting the metabolic flux towards either flavonoids or lignin production, thus further research must prioritize validating the roles attributed to lignin and flavonoids in melon's defense against P. cubensis through targeted gene editing or overexpression strategies targeting the HCT gene (Discussion section, lines 419-422).

Comments

  1. Line 41, please cite publications as recommended by the Editor.

     Thank you! We have modified it as recommended by the Editor in lines 48           and lines 579.

  1. Please provide details of the manufacturer of the statistical software used to analyse the study data.

     Thanks for pointing this out. We have added all manufacturer details in lines         455, 457, 461, 474, 505 of revised manuscript currently.

  1. References

Please remove publications older than 10 years.

 We thank the reviewer for this suggestion. We have deleted all publications older than 10 years, except for a publication from 2001 on gene expression detection (Livak, K.J.; Schmittgen, T. Analysis of relative gene expression data using real-time quantitative PCR and the 2-DDCt Method. Methods. 2001, 25, 402-408g). This publication has been cited over 160,000 times and is considered the gold standard for gene expression detection.
